# Causes for increased flood frequency in central Europe in the 19th century

Stefan Brönnimann,[1,2,*] Luca Frigerio,[1,2] Mikhaël Schwander,[1,2,3] Marco Rohrer,[1,2] Peter Stucki,[1,2] Jörg Franke[1,2]

[1] Oeschger Centre for Climate Change Research, University of Bern, Switzerland

[2] Institute of Geography, University of Bern, Switzerland

[3] Federal Office of Meteorology and Climatology MeteoSwiss

*correspondence to: stefan.broennimann@giub.unibe.ch*

## Abstract

Historians and historical climatologists have long pointed to an increased flood frequency in Central Europe in the mid and late 19[th] century. However, the causes have remained unclear. Here, we investigate the changes in flood frequency in Switzerland based on long time series of discharge and lake levels, of precipitation and weather types, and based on climate model simulations, focusing on the warm season. Annual series of peak discharge or maximum lake level, in agreement with previous studies, display increased frequency of floods in the mid 19th century and decreased frequency after the Second World War. Annual series of warm-season mean precipitation and high percentiles of 3-day precipitation totals (partly) reflect these changes. A daily weather type classification since 1763 is used to construct flood probability indices for the catchments of the Rhine in Basel and the outflow of Lake Lugano, Ponte Tresa. The indices indicate an increased frequency of flood-prone weather types in the mid 19[th] century and a decreased frequency in the post-war period, consistent with a climate reconstruction that shows increased (decreased) cyclonic flow over Western Europe in the former (latter) period. To assess the driving factors of the detected circulation changes, we analyse weather types and precipitation in a large ensemble of atmospheric model simulations driven with observed sea-surface temperatures. In the simulations, we do not find an increase in flood-prone weather types in the Rhine catchment in the 19th century, but a decrease in the post-war period that could have been related to sea-surface temperature anomalies.

## 1. Introduction

Floods are some of the costliest natural hazards in Europe (EEA, 2018). In typical pluvio-nival river regimes in Central Europe, floods are often triggered by one or several days of heavy precipitation, but some rivers also exhibit winter floods due to longer periods of large-scale precipitation or spring floods due to heavy precipitation, amplified by snow melt. Such factors might change in the future. For instance, heavy precipitation events will become more

intense in the future according to global climate model simulations (Fischer and Knutti,
2016). An intensification of heavy precipitation events is also found in regional model
simulations for Europe north of the Mediterranean (Rajczak and Schär, 2017). With
increasing temperature, snow melt occurs earlier in the year, changing river regimes.
Furthermore, also precipitation extremes might shift seasonally (Brönnimann et al., 2018;
Marelle et al., 2018). While changes in seasonality have been found for European floods
(Blöschl et al., 2017), no general increase in flood frequency has so far been detected
(Madsen et al., 2014). However, past records suggest that there is considerable decadal
variation in flood frequency (e.g., Sturm et al., 2001; Wanner et al., 2004; Glaser et al., 2010).
It is reasonable to assume that such variations will continue into the future. In this paper we
focus on decadal variability during the past 200 years.
An increased flood frequency in the 19th century was already perceived by contemporary
scientists across central Europe and affected the political debates on deforestation as a
potential cause (e.g., Brückner, 1990). The changing frequencies of flood events in Central
Europe over the past centuries have been analysed in detail during the past 20 years (e.g.,
Mudelsee et al., 2004; Glaser et al., 2010). One result is that different river basins behave
differently due to different hydrological regimes and different seasonality of floods. For
instance, Glaser et al. (2010) found a prominent phase of increased flood frequency in central
European rivers from 1780 to 1840, but mainly in winter and spring. This may not apply to
Alpine rivers, which are more prone to floods in summer and autumn. Periods of increased
flood frequency have also been analysed with respect to reconstructions of atmospheric
circulation (e.g., Jacobeit et al., 2003; Mudelsee et al., 2004). Jacobeit et al. (2003) find that
the large-scale zonal mode, which characterizes flood events in the 20[th] century, does not
similarly characterize flood-rich periods during the Little Ice Age (their analysis, however,
does not cover the 19[th] century). For summer floods, Mudelsee et al. (2004) find a weak but
significant relation to meridional airflow. Quinn and Wilby (2013) were able to reconstruct
large-scale flood risk in Britain from a series of daily weather types back to 1871 and found
decadal scale changes in circulation types.
For several catchments in the Alps and central Europe, studies have suggested an increased
frequency of flood events in the mid 19th century (Pfister 1984, 1999, 2009; Stucki and
Luterbacher, 2010; Schmocker-Fakel and Naef, 2010a,b; Wetter et al., 2011). However, the
causes of this increased flood frequency remain unclear. Besides human interventions such as
deforestation or undesigned effects from water flow regulations (Pfister and Brändli, 1999;
Summermatter, 2005), this includes the role of cold or warm periods and changes in
atmospheric circulation. Proxy-based studies, though focussing on longer time scales, find
that in the Alps, cold periods were mostly more flood prone than warm periods (Stewart et al.,
2011; Glur et al., 2013); the last of these cold and flood prone periods in the latter study is the
19th century. Glur et al. (2013) relate periods of increased flood frequency in the past 2500
years to periods of a weak and southward shifted Azores High. Even more remote factors
could have played a role. Using climate model simulations, Bichet et al. (2014) investigated
the roles of aerosols and of remote Pacific influences on precipitation, albeit focusing on the
seasonal mean. Finally, Stucki et al. (2012) performed case studies of the strongest 24 flood
events of the last 160 years. They characterised five flood-conducing weather patterns,
although each extreme event had its individual combination of contributing factors.
In our paper, we aim to combine analyses of daily weather, reconstructions, and climate
model simulations to elucidate potential causes leading to an increased flood frequency in
Switzerland. While previous studies have focused on monthly or seasonal reconstructions, or
on individual cases, we study the daily weather back to the 18th century in a statistical
manner, thus bridging the gap between event analyses and paleo-climatological studies.
In this study we track the flood-frequency signal from historian documents to observations
and simulations. Using long data series on floods (discharge and lake level), precipitation,
daily weather types, and climate model simulations, we investigate whether an increased
frequency of flood events was due to a change in seasonal mean or extreme precipitation and
whether this can be related to change in weather conditions. We also address the underlying
hydro-meteorological and climatological causes in model simulations. The paper is organised
as follows. Section 2 describes the data and methods used. Section 3 describes the results. A
discussion is provided in Section 4. Conclusions are drawn in Section 5.

**2. Data and Methods**
*2.1. Discharge data*
For the analysis of the flood frequency, we used annual peak discharge measurements from
Basel, Switzerland, since 1808 (Wetter et al., 2011) as well as annual peak lake level data for
Lake Constance, Constance (since 1817, supplied by the German Landesanstalt für Umwelt,
Messungen und Naturschutz Baden-Württemberg) and Lago Maggiore, Locarno (Locarno
(Swiss Federal Office for the Environment FOEN) since 1868. The Lago Maggiore data were
corroborated by instrumental measurements at Sesto Calende for past floods since 1829 (Di
Bella, 2005) and by reconstructed lake levels for floods prior to that time both for Locarno
and Sesto Calende (Stucki and Luterbacher, 2010). Further, we used a daily discharge time
series for Basel and Ponte Tresa, Ticino, since 1901 from the FOEN. Figure 1 gives an
overview of the catchments and locations used in this paper; Figure 2 shows the series.
Some of the series have potential inhomogeneities. Major corrections in the catchments or
lakes were carried out in 1877 (Jura Waters correction, affecting the Aare and thus the Rhine),
between 1888 and 1912 (Ticino in the Magadino plain), and 1943 (regulation of the level of
Lago Maggiore). Lake Constance was and still is unregulated, but Jöhnk et al. (2004) argue
that the level decreased by 15 cm between 1940 and 1999 due to upstream reservoirs. Based
on model simulations, Wetter et al. (2011) estimate that the Jura Waters correction led to a
reduction of peak discharges in Basel by 500 to 630 $m^3$/s. A further possible inhomogeneity
concerns the level of Lago Maggiore. The flood of 1868 reportedly has led to erosion at the
outflow, lowering the peak lake levels after the event. We will address these issues in Sect. 3.
Note that in terms of underlying processes, lake floods slightly differ from river floods. They
depend on the antecedent lake level, which carries a longer memory with it.

*2.2. Precipitation data*
Unfortunately, hardly any daily precipitation series covers the entire, approx. 200- year period
considered here. The only long series in Switzerland is from Geneva (Füllemann et al., 2011),
with daily precipitation data reaching back to 1796. Note that this series has not been
homogenized prior to 1864, and that it might not be representative for the northern side of the
Alps. Much more daily records exist from Switzerland from 1864 onward, the start of the
Swiss network. We use data for Lugano (Ponte Tresa catchment), as well as from a number of
other stations (Affoltern, Basel, Altstätten, Bellinzona, Lohn, Engelberg, see Fig. 1). Monthly
precipitation was taken from the gridded HISTALP data set (Hiebl et al., 2009).
Earlier studies (e.g., Glaser et al., 2010; Stucki et al., 2012) indicate that in the region of
interest, most floods occur in the warm season (hereafter May to October). The only notable
exception is the Christmas flood of 1882 (marked by a star in Fig. 2). In this paper, we
therefore show the results only for the warm season. From both daily precipitation series we
calculate the maximum precipitation amount over 3 days per warm season, denoted Rx3day.
From the gridded HISTALP data set we calculated warm season precipitation averages for
two regions (Fig. 1): A region north [46.5-47.5°N, 6.5-10°E] and a region south [45.75-
46.25°N, 8.5-9.25°E] of the Alpine divide.

*2.3. Weather type reconstruction*
In order to address flood-inducing weather patterns, we use the daily weather type
reconstruction for Switzerland by Schwander et al. (2017), which reaches as far back as 1763.
The weather types used in this paper are an extension of the CAP9 weather types of
MeteoSwiss (Weusthoff, 2011) into the past, using station data and classifying each day
according to its Mahalanobis distance from the centroids of the weather types in the
calibration period. However, as two of the types were not well discernible from two other
types, the respective types were merged such that only seven types remain (CAP7, see
Schwander et al., 2017). This assures a good quality of the reconstruction. After 1810, the
probability of each day to be attributed to the right class is higher than 80%, after 1860 it is
higher than 85% (see Schwander et al. 2017). Figure 3 shows the averages of sea-level
pressure per CAP7 weather type.

*2.3. Reanalyses*
To corroborate our results, we also consulted the "Twentieth Century Reanalysis" version 2c
(20CRv2c, Compo et al. 2011). Specifically, we used daily data of precipitation, precipitable
water (PWAT), and $u$ wind at 850 hPa for the grid point 6°E/48°N, representing the Basel
catchment. From these data we calculated Rx3d as well as a $u_{850hPa}$*PWAT as a measure of
moisture transport from the west towards the Alps. This is important as so-called
"atmospheric rivers" are important precursors to Alpine flood events (Froidevaux and
Martius, 2016). We also calculated CAP7 weather types from 20CRv2c as described in
Rohrer et al. (2018). In brief, we attributed each day to the closest circulation type centroid
according to its Euclidian distance. Centroid were defined in the 1957-2010 based on the
MeteoSwiss classification (Weusthoff, 2011). Note that all calculation were performed for
each of the 56 members of 20CRv2c individually.

*2.5. Climate model simulations and reconstructions*
For the analysis of atmospheric circulation during the 19th and 20th century, we use the
reconstruction EKF400 (Reconstruction by Ensemble Kalman Fitting over 400 years, Franke
et al., 2017). This global, three-dimensional reconstruction is based on an off-line data
assimilation approach of early instrumental, documentary and proxy data into an ensemble of
climate model simulations. It provides an ensemble of 30 monthly reconstructions back to
1600. Here we use the ensemble mean and analyse geopotential height (GPH) and vertical
velocity at 500 hPa, wind at 850 hPa as well as precipitation.
Finally, we compare the observations-based results with a large ensemble of climate model
simulations. We use a 30-member ensemble of atmospheric simulations performed with
ECHAM5.4 (T63) termed CCC400 (Chemical Climate Change over 400 years), which is the
set of simulations that also underlies EKF400. The simulations cover the period 1600 to 2005
and are described in Bhend et al. (2012). Their most important boundary conditions are sea-
surface temperature (SST) data by Mann et al. (2009). From these SSTs we also calculated
indices of the Atlantic Multidecadal Oscillation (AMO) and the Pacific Decadal Oscillation
(PDO) following the definitions by Trenberth and Shea (2006) and Mantua et al. (1997),
respectively (see Brönnimann, 2015, for extensive comparisons of these indices and CCC400
results). Note that in these simulations, the long-term changes in land-surface properties were
misspecified. We therefore performed an additional simulation with corrected land surface to
assess the impacts (Rohrer et al., 2018). While no impacts were found in heavy precipitation
and weather types, warm-season average precipitation showed a too strong drying trend,
which we adjusted to match that of the corrected simulation. In any case, the discrepancy
concerns the long-term change and not decadal-to-multidecadal variability.
Similar as for 20CRv2c, we use daily precipitation representative of the Aare catchment
(47.5° N/7.4° E, see Brönnimann et al., 2018) and the CAP7 weather types from CCC400.
The CAP7 weather types were evaluated by Rohrer et al. (2018): Although the model shows a
zonal bias (too frequent westerly types), the decadal variability of weather type frequencies
within the simulations may give some indications as to possible contribution due to SST
anomalies or external forcings.

*2.6. Construction of a flood probability index*
From the weather types described above, we construct a flood probability index (FPI) for each
river catchment following the basic methodology of Quinn and Wilby (2013). The FPI weighs
the frequency of weather types according to their flood-proneness. To determine the weights,
we used daily discharge data during the period 1901-2009 for Basel and Ponte Tresa. Flood
events were defined using a peak-over-threshold approach. The 98[th] percentile of warm
season days was taken as a threshold, and a declustering was applied by combining events
with a maximum distance of 3 days. Compositing the events around the day of maximum
discharge showed enhanced discharge already several days prior to the event. Therefore, we
also considered weather types on the five days prior to the event (Froidevaux (2014),
analysing the role of antecedent precipitation for floods in Swiss rivers, find a somewhat
shorter interval, but analysed smaller catchments). In the following we analyse the weather
types during on flood events and the preceding 7 days.
Figure 4 (top) shows the frequency of weather types during all warm season days for the
period 1901-2000. The types „northeast, indifferent" and „west-southwest, cyclonic", and
„east, indifferent" make up 60% of all days. The most rare weather type „high pressure"
accounts for 5% of all days. The middle and bottom panels show the fraction of flood events
per weather type for Basel and Ponte Tresa (dividing the fractions in the bottom panels series
by the frequencies in the top panel yields $w_{tl}$). Of all flood days in Basel, 60% are either
„northeast indifferent" or „north cyclonic" types. The two days prior to the event are
dominated (77%) by the three "cyclonic" types, and an increase of cyclonic types is even
found five days ahead of the flood event (65% versus 42% on average). For Ponte Tresa, type
7 („westerly over southern Europe, cyclonic") is the most flood prone, followed by „west-
southwest cyclonic". The former dominates particularly one to five days ahead of the event.
On these days, type 7 is 4 times more frequent than on average.
A seasonal or annual flood probability index $FPI_y$ can be defined in the following way. For all
event days in our calibration period 1901-2009 (and similarly for preceding days, $l$ indicates
the lag and ranges from -5 to 0), we analysed the absolute frequency of a given weather type $t$
($n_t$) relative to all event days ($n_l$) and divided this by the absolute frequency of that weather
type on all days ($n_t$). This ratio was termed $w_{tl}$:
$$w_{tl} = \frac{n_{tl}/n_l}{n_t} \qquad\qquad (1)$$
To determine the $FPI$ for a given year $y$ (in our case, a warm season) we analysed the absolute
weather type frequencies in that year (warm season), $n_{ty}$, and multiplied it with the
corresponding weights $w_{tl}$ for a given lag $l$. This results in one time series for each lag $l$. The
four series were then combined to provide the index $FPI_y$ using a weighted average with
weights $v_l$:
$$FPI_y = \sum_l v_l \sum_t n_{ty} w_{tl} \qquad\qquad (2)$$
Based on the results of Figure 4, the weights ($v_l$) for days -5 to 0 were set to 1/16, 1/8, 1/4,
1/4, and 1/8, respectively (assigning equal weights or using a shorter window yields very
similar results). Note that weights were recalculated for the $FPI$ from 20CRv2c.
Quinn and Wilby (2013) used annual frequencies of the weather types to define the FPI. Here
we calculated a daily index $FPI_d$, which (unlike the annual index) takes the actual sequence of
weather types into account, such as during the passage of a cyclone. Equation (2) can be used
for the daily index, with the same weights $v_l$ and $w_{tl}$ as for $FPI_y$, but the frequency $n_{tdl}$ is now
either zero or one:
$$FPI_d = \sum_l v_l \sum_t n_{tdl} w_{tl} \tag{3}$$
The result is a daily index $FPI_d$ whose warm season average is by definition equal to $FPI_y$, but
which allows also studying other statistics. To test the daily index for the case of Basel, we
studied composites of $FPI_d$, average daily precipitation from all sites North of the Alps
(Affoltern, Altstätten, Basel, Engelberg, Geneva, and Lohn), moisture transport $u_{850}$*PWAT
from 20CRv2c, and discharge in Basel for two types of composites: (1) for peak-over-
threshold flood events and (2) for peak-over-threshold events of $FPI_d$ (defined in the same
way, i.e., as declustered 98$^{th}$ percentile). As expected, flood events are related to a clearly
increased $FPI_d$ (Fig. 5, left). The average reaches 1.67, which means a 67% increase in flood
probability. This corresponds to the 83$^{rd}$ percentile of $FPI_d$. Moisture transport is increased (to
its 75$^{th}$ percentile) 5 to 2 days prior to the flood event. Precipitation reaches its 97$^{th}$ percentile
on days 1 and 2 prior to the event. The mean of the selected flood events corresponds to a
quantile of 99.4%. Compositing the same variables for instances with a high $FPI_d$ (Fig. 5,
right), we find similarly high percentile (99.3%) for the mean of the selcted $FPI_d$ events. We
also find high moisture transport (79$^{th}$ percentile) and precipitation (89$^{th}$ percentile) two days
ahead of the event. The $FPI_d$ clearly captures the passage of active cycones. Discharge in
Basel is also increased, but only to its 71$^{st}$ percentile.
Thus, the index captures flood events and also moisture transport and precipitation well,
although with a high rate of „false alarms" (i.e., not all $FPI_d$ events lead to floods). This can
be expected for such a coarse classification. Classifications with more types were also
reconstructed, but less skilfully and hence we prefer CAP7 (Schwander et al. 2017). Another
cause are the preconditions for flood events, particularly for such a large catchment as the
Rhine. Discharge in Basel is on average above its 75$^{th}$ percentile already a week or more prior
to the event, perhaps due to the passage of previous cyclones (not captured in $FPI_d$). A third
cause for false alarms is the different sample size of flood events ($n = 110$) and „FPI events"
($n = 285$) despite using the the same threshold and declustering. This is due to the different
temporal structure of the time series. Two thirds of the $FPI_d$ events cannot be floods even if
the match was perfect.
High percentiles of $FPI_d$ are thus not suitable for studies of interannual-to-decadal variability.
Flood-conducive cyclone passages occur almost every summer and hence high percentiles of
$FPI_d$ show little interannual variability. We use the warm season 75[th] percentile to capture the
upper part of the distribution as well as the 50th percentile and the mean (i.e., $FPI_y$) to capture
the central tendency

**3.   Results**
*3.1. Flood frequency*
To begin with, we analysed the flood series in order to test whether the reported increased
flood frequency in the mid-19th century is also found in our series (Fig. 4). The first thing we
note is that floods do not occur synchronously across the considered catchments. The same is
true for annual peak discharge series in general, as evidenced by low Spearman correlations.
For instance, the series for the Rhine in Basel is uncorrelated with the series of Lago
Maggiore and only moderately (coefficient of 0.36) with the series of Lake Constance, even
though the latter comprises a large sub-catchment.
Was flood frequency higher in the mid-19[th] century? In fact, each series exhibits prominent
peaks in the 19th century, e.g., the Rhine in Basel in 1817, 1852, 1876, 1881, and 1882 (see
Stucki et al., 2012), Lake Constance in 1817 (see Rössler and Brönnimann, 2018) and Lago
Maggiore in 1868 (Stucki et al. 2018). However, we also note a period of low flood frequency
in Basel from the 1920s to 1970s, in agreement with a low frequency of peak-over-threshold
events in Basel and Ponte Tresa. For further analyses we defined the 30-yr periods with
highest and lowest flood frequencies, respectively, as follows: From the annual series we
defined floods as exceedances of the 95[th] percentile of the 1901-2000 period (dashed line).
Note that even accounting for a shift of 630 m$^3$/s due to the Jura Waters correction would not
change the selected events of the Rhine in Basel, neither would a correction for a linear 15 cm
trend of Lake Constance after 1940 due to an increasing number of water reservoirs upstream
(cf. Jöhnk et al., 2004). However, the inhomogeneity caused by the 1868 event might be
substantial. We therefore considered pre-1868 data only qualitatively.
Counting annual floods in all series as well as counting the daily peak-over-threshold events
for Basel and Ponte Tresa both yields the same 30-yr period with lowest flood frequency:
1943-1972. The period with highest flood frequency is only defined by counting annual
floods. Not including pre-1868 Lago Maggiore data, the period 1847-1876 is the most flood-
rich. This is further supported by the historical data for Lago Maggiore, which suggest
additional strong flood events in that period. However, earlier 30-yr periods might be equally
or even more flood-rich, according to reconstructed flood events.
In the following we assess differences in a variable in each period relative to a corresponding
climatology (a sample consisting of 30 yrs before and 30 yrs after the period to further reduce
the effect of centennial-scale changes) as well as between the two periods with a Wilcoxon
test.

*3.2. Precipitation*
In a second step, we analysed warm-season mean precipitation and Rx3day for the regions
north or south of the Alps (Figs. 6 and 7). In both regions, warm season precipitation is
correlated significantly (Spearman correlation of 0.45 and 0.50, respectively) with annual
maximum discharge, clearly indicating that the floods under study are caused by excess
precipitation. In both regions, precipitation was slightly above the 20th century mean (dashed)
during most of the 19th century and below average during the flood-poor period. The
difference between the flood-rich (1847-1876) and the flood-poor (1943-1972) periods is
significant (p-value of the Wilcoxon test: $p = 0.027$) for the Ponte Tresa catchment. For the
Basel catchment, both periods deviate significantly negatively from the corresponding
neighbouring decades ($p = 0.049$ and $0.030$ for the flood-rich and flood-poor period,
respectively), which is unexpected for the flood-rich period. Their difference is not
significant.
Rx3day for Geneva and Lugano are shown exemplarily to assess the role of extreme
precipitation. For Geneva, we find two pronounced extremes (1827, 1888), both of which
were discussed in newspapers (NN, 1827) and thus are considered real. For both stations, the
decreased intensity of Rx3d in the flood-poor period relative to neighbouring decades is
significant ($p = 0.026$ and $0.038$ for the Rhine and Ponte Tresa catchments, respectively). A
similar decrease at the same time is also found for other series in Switzerland (Fig. 8 shows
six long series). Calculating for each series the annual exceedance frequency of the 95th
percentile (based on the 1901-2000 interval) of Rx3d and then averaging over all 8 series
shown in Figs. 6 to 8, we obtain a time series of the ratio of stations exceeding their 95th
percentile in a given year. This series shows lower values in the 1943-1972 period than in the
following 30 year period and even lower than in the late 19th century.
In Section 3.1 we found clear changes in flood frequency. This section shows that at least the
flood-poor period was related to a reduction in the precipitation amount and intensity of Rx3d
events, while results for the flood-rich period are ambiguous.

*3.3. Moisture transport*
In addition, we consider moisture transport from the West towards the Alps, which we
analyse in 20CRv2c for the Basel catchment. As a diagnostic we calculate, similar to Rx3d,
the largest 3-day average of $u_{850hPa}$*PWAT per summer season. This proxy for westerly
moisture transport is shown together with Rx3d and $FPI_y$ (both also calculated from
20CRv2c) in Fig. 9. For Rx3d and $FPI_y$ we also show the observations-based series.
Results for the flood-rich period are ambiguous, and discrepancies to the observations-based
series are large in parts, as is seen in $FPI_d$ in the 1850s and 1880s in Fig. 9. This may be
explained by the fact that 20CRv2c is prone to errors in the early decades (see Rohrer et al.
2018). Agreement between observations and 20CRv2c increases after 1900. Specifically,
moisture transport shows similar decadal variability as *FPI* or precipitation, with higher
values prior to 1940 and lower values afterwards. Although 20CRv2c alone does not permit
the interpretation of decadal changes, we note that the changes are fully consistent with those
in our independent time series.

*3.4. Weather and large-scale flow*
In the next step step, we analyse the link of flood events to atmospheric circulation and its
multidecadal changes by means of the $FPI_d$ statistics (see Sect. 2.5). The temporal
development for Basel (Fig. 6, bottom) and Ponte Tresa (Fig. 7, bottom) is similar for all
indicators (mean, median or 75[th] percentile), and the Spearman correlations of the Basel $FPI_d$
series with the annual maximum discharge at Basel are statistically significant (p = 0.005 to
0.014). This shows that the *FPI* is a good predictor for flood variability.
For both catchments, the indices reveal clear multidecadal variability. Indices are generally
positive from the 1810s to 1900s (with a secondary maximum in the 1920s and 1930s) and
negative from the 1940s to around the 2000s. Both periods are longer than those selected in
our study. The differences in the $FPI_d$ between our flood-rich and flood-poor period is
significant in both catchments for all three indices (max. p-value is 0.0023). The flood-rich
period does not differ significantly from the neighbouring decades (which also show high
values of the FPI) in any of the indices, whereas the flood-poor period shows lower values
than the neighbouring decades (p = 0.047 and 0.067 for Basel and Ponte Tresa, respectively).
From these analyses we can conclude that the change in precipitation amount and intensity
found in the previous Section was related to the FPI. The flood-rich and flood-poor periods
clearly differ with respect to occurrence of weather types, i.e. large-scale atmospheric flow.
Floods are extreme and thus rare events, but the causes for changes in extremes do not need to
be rare. Changes in extremes may be the expression of a shift in the underlying distribution.
For instance, the correlations of the 75$^{th}$ and 90$^{th}$ percentile of $FPI_d$ with the mean are 0.92
and 0.77 for the Basel catchment and 0.95 and 0.91 for the Ponte Tresa catchment.
Additionally, for the case of floods, Fig. 5 shows that preconditions (and thus the previous
cyclone) matter. We therefore analyse to what extent the change in weather types is mirrored
in the multi-decadal atmospheric circulation statistics.
We analysed the two periods in global climate reconstructions (EKF400), each relative to its
climatology as well as the difference between the two (Fig. 10). The anomalies for the flood-
rich period show clear negative GPH anomalies over western Europe and strengthened flow
from the north-west. The extension of the Azores onto the European continent weakened. This
pattern becomes a lot stronger and clearer when contrasting the two periods (flood-rich minus
food-poor). The anomalies for the flood-poor period show strengthened high-pressure
influence over Central Europe, descent, and dryness with anomalous flow from the north east.
In all, the large-scale analysis confirms the results from the *FPI*: It shows clearly that the shift
in weather types was an expression of multi-decadal variability of atmospheric circulation
over the full North Atlantic-European sector, consisting of a more zonal and southward-
shifted circulation.

*3.5. Climate model simulations*
We have seen that the decadal-to-multidecadal changes in flood frequency can be related to
changes in weather types, which are part of large-scale flow anomalies. In the fourth step, we
analysed whether this can in turn be attributed to influences such as sea-surface temperature
variability modes as depicted by atmospheric model simulations (CCC400) or whether the
decadal-to-multidecadal changes are due to random, possibly atmospheric variability.
Concretely, we analysed warm-season mean precipitation and Rx3d for a grid point north of
the Alps and calculated $FPI_d$ and its statistics for each member. We then averaged the results
across all 30 CCC400 members (one corrupt member was excluded for $FPI_y$). This is
meaningful because changes in the ensemble mean reflect a common signal which must be
due to the common boundary conditions of the simulation. Figure 11 shows the series in a
smoothed form (31-yr moving average) for visualisation.
Indeed, we note that the agreement between modelled and observation-based FPI is not good
in the 19[th] century; the broad 19[th] century peak in the observation-based FPI is missing in the
model. In addition, the analysis reveals downward trends in mean precipitation (although the
series is trend-corrected) as well as in Rx3d. Quantitatively, the trend in mean precipitation
amounts to -1.88% per century, which is rather small (much smaller than in the observations).
Due to this trend it is not surprising that significant differences in seasonal mean precipitation
appear between the two periods which may be unrelated to decadal-to-multidecadal variability
but rather to multi-centennial trends. Differences between the averages of the flood-rich and
the flood-poor periods across the ensemble are not significant for Rx3d and around the
significance limit for $FPI_y$ (Wilcoxon test: $p = 0.043$).
In the model, the flood-rich period is not significantly different from neighbouring decades in
any of the measures, but the flood-poor period appears as a potentially flood-poor period in
seasonal mean precipitation and $FPI_y$ (Wilcoxon test: $p = 0.013$ and $p = 0.004$, respectively).
Only model boundary conditions can explain this, and the arguably dominant contribution is
from SSTs. Among the well-known SST variability modes, it is in fact the PDO index that
explains the $FPI_y$ most successfully. However, the Spearman correlation remains low and not
significant in view of the low number of degrees of freedom, even after detrending.
We infer from these analyses that our climate model simulations do not reproduce the flood-
rich period, but the flood-poor period appears as a feature.

**4. Discussion**
While tracking the flood-frequency signal, we have found a number of links and
dependencies; these are discussed in the following. For instance, previous studies found an
increased flood frequency in Switzerland in the 19[th] century (Pfister 1984, 1999, 2009; Stucki
and Luterbacher, 2010; Schmocker-Fakel and Naef, 2010a,b; Wetter et al., 2011) as well as a
decrease in the mid 20th century, sometimes referred to as the „disaster gap" (Pfister, 2009;
Wetter et al., 2011). The series used in this paper confirm the general tendency. Schmocker-
Fackel and Naef (2010a,b) identify 1820-1940 as a flood-rich period, while we use much a
shorter period. However, our *FPI* is consistent with a longer flood-rich period around 1820-
1940, i.e., the difference between 1820-1940 and 1943-1972 is also highly significant ($p < 0.00001$).
425 0.00001).

Rx3day series from Geneva and Lugano together with series from a larger number of Swiss
stations confirm a multidecadal period around the 1960s with reduced intensity of Rx3d. The
change in the frequency of floods, which are rare events, is related to a change in mean
climate. For instance, warm season mean precipitation shows changes that are concurrent with
those of flood frequency, with significant correlations. We also find consistent changes for
high percentiles of the *FPI* and its mean.
Schmocker-Fackel and Näf (2010a) analysed the relation between floods and weather types
for the period after 1945 and manual assignments based on weather reports before that year.
Here we can make use of a new, daily 250-yr weather type reconstruction. As in Schmocker-
Fackel and Näf (2010a), we find that events south of the Alps and those north of the Alps are
related to slightly different weather type characteristics, although indices for both regions are
highly correlated on all time scales. Our *FPI* shows clear multidecadal variability, with high
values during most of the 19th century and a secondary peak in the 1920s and 1930s, and
lower than average values in the post-war period. After around 1980, the FPI returned to
average values. The *FPI* reflects passing cyclones, but it also captures episodes of strong
moisture transport, and in fact annual 3-day maxima of moisture transport in 20CRv2c show
similar multidecadal variability.
In agreement with Schmocker-Fackel and Näf (2010a,b), we find no imprint on the classical
NAO pattern and also no clear weakening of the Azores high during the flood-rich period.
However, we find that the extension of the Azores high onto the European continent
weakened, and we find clear negative GPH anomalies over western Europe, strengthened
north-westerly advection, and large-scale ascent. This indicates a more zonal, southward-
shifted circulation over the North Atlantic-European sector during the flood-rich period.
Opposite anomalies, i.e., positive GPH anomalies and descent, are found for the flood-poor
period, which was in fact associated with heatwaves and strong droughts in Central Europe.
Brugnara and Maugeri (2019), find a regime shift in total precipitation and wet-day frequency
for a southern region of the Alps, and for a period after the 1940s which coincides with the
flood-poor period.
The flood-poor period might carry imprints of oceanic influences. Sutton and Hodson (2005)
related summer climate anomalies on both sides of the Atlantic in the wider 1931-1960 period
to changes in the AMO. We do not find a significant correlation between our flood and
precipitation indicators and the AMO; a possible relation to the PDO index is possible but not
confirmed. The flood-poor period partly overlaps with a period of poleward displacement of
the northern tropical belt, which is understood to be caused by sea-surface temperature
anomalies and is reproduced in climate models (Brönnimann et al., 2015). Our EKF400
analysis is thus consistent with the results of the latter study.

**5. Conclusions**
Flood frequency in Central Europe exhibits multidecadal changes, which has been
demonstrated based on historical records. The causes for the increased flood frequency in
Switzerland in the 19th century as well as for the decreased flood frequency around the mid-
20th century are long-standing issues. In this study we have tracked these changes from flood
records through precipitation records, weather type statistics and large-scale circulation
reconstructions all the way to oceanic influences as expressed in atmospheric model
simulations. The change in flood frequency is arguably the expression of changes in mean
climate. We attribute the changes in flood frequency to changes in mean precipitation and in
the intensity of Rx3d. In turn, these are related to a change in cyclonic weather types over
Central Europe. These changes indicate a shift in large-scale atmospheric circulation, with a
more zonal, southward shifted circulation during the flood-rich period relative to the flood
poor period. Precipitation and circulation changes are only to a small part reproduced in
climate model simulations driven by observed sea-surface temperatures, which points to
random atmospheric variability as an important and complementary cause.
The analyses show that decadal variability in flood frequency occurred in the past; and is
likely to continue into the future. Better understanding its relation to weather regimes, large-
scale circulation, and possibly sea-surface temperature may help to better assess seasonal
forecasts and projections. Finally, the study also shows that the Quinn and Wilby (2013)
methodology also works for flood risk in Switzerland.

*Acknowledgements:* This work was supported by Swiss National Science Foundation projects
RE-USE (162668), EXTRA-LARGE (143219), and CHIMES (169676), by the European
Commission (ERC Grant PALAEO-RA, 787574) and by the Oeschger Centre for Climate
Change Research. Simulations were performed at the Swiss National Supercomputing Centre
CSCS.

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

 **Figures**

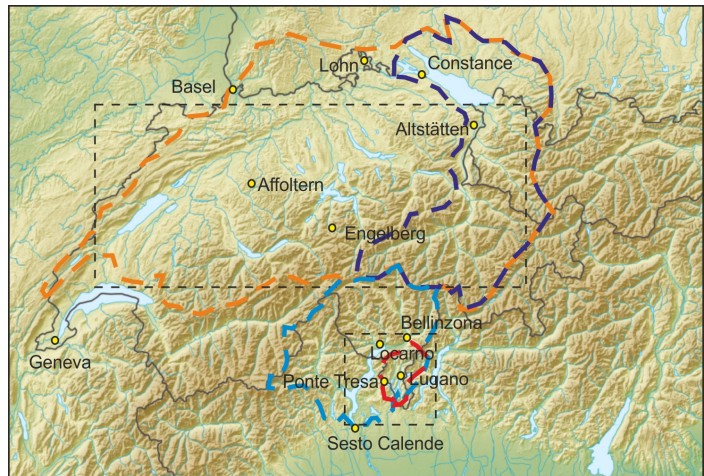


**Fig. 1.** Topographic map of the central Alps showing the catchments and locations mentioned in the
text, the catchments of the Rhine in Basel (orange), Lake Constance (dark blue), Lago Maggiore (light
blue) and Ponte Tresa (red). The rectangle boxes indicate the areas chosen for averaging precipitation
in the HISTALP data.

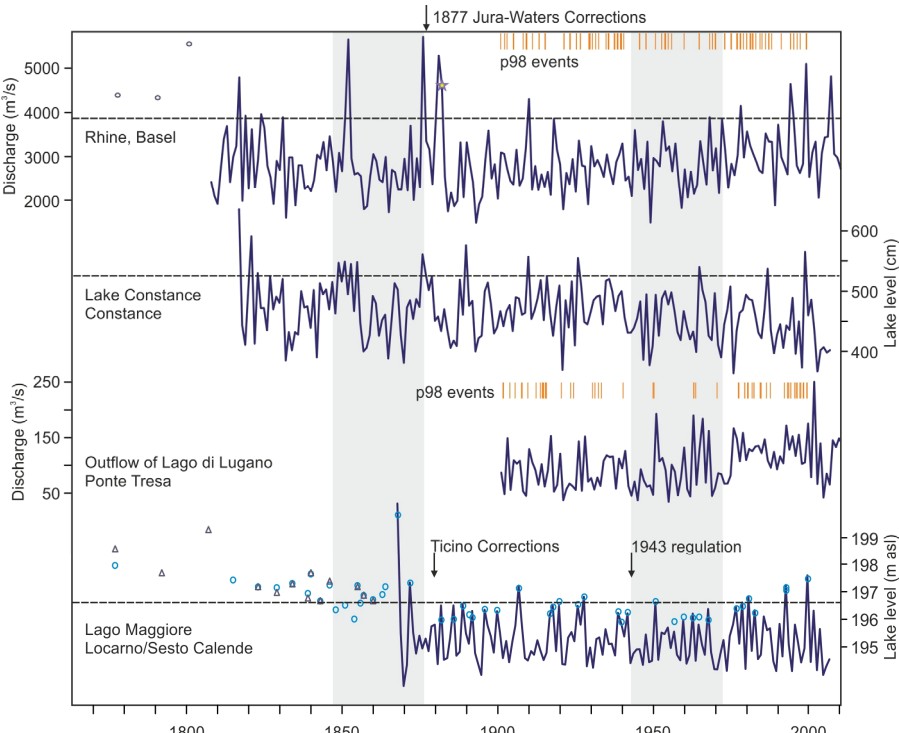


**Fig. 2.** Time series of annual maxima of discharge or lake level in four catchments. Symbols denote
reconstructed floods based on historical sources (circles for Rhine, Basel, from Wetter et al. 2011,
Triangles for Lago Maggiore, Locarno, from Stucki and Luterbacher, 2010, light blue circles for Lago
Maggiore refer to floods at Sesto Calende according to Di Bella, 2005, from reconstruction prior to
1829 and measurements afterwards, adjusted to Locarno by adding the average difference between the
two during floods after 1868, i.e., 0.49 m). Arrows indicate major river corrections. Orange bars
indicate the peak-over threshold events in the 1901-2000 period that were used to calibrate the FPI.
Grey shading denotes the flood-rich period (1847-1876) and flood-poor period (1943-1972). Dashed
lines indicate the 95[th] percentile from 1901-2000. The star marks the Christmas flood of 1882.

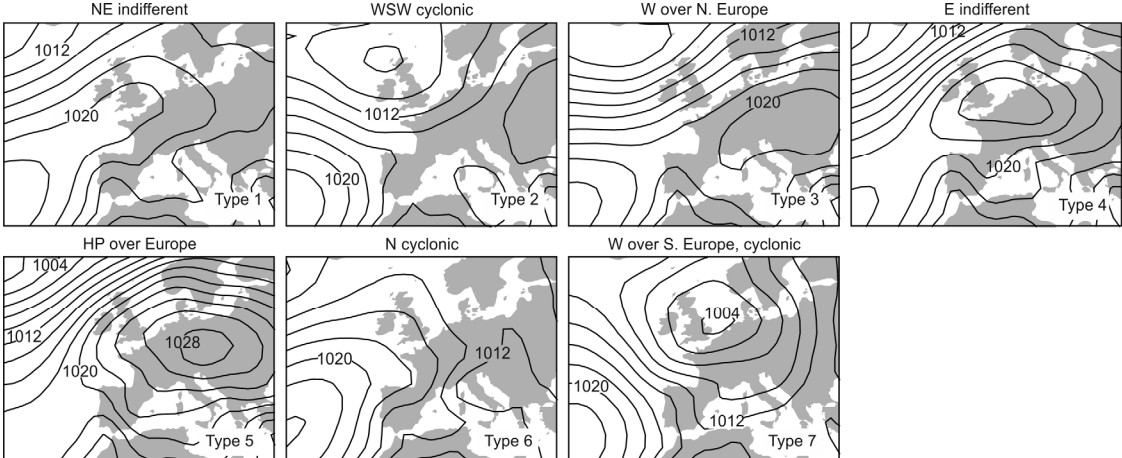


**Fig. 3.** Sea-level pressure averaged for each of the 7 weather types in CAP7 over the warm season (May-Oct) for the period 1958-1998 based on 20CRv2c.


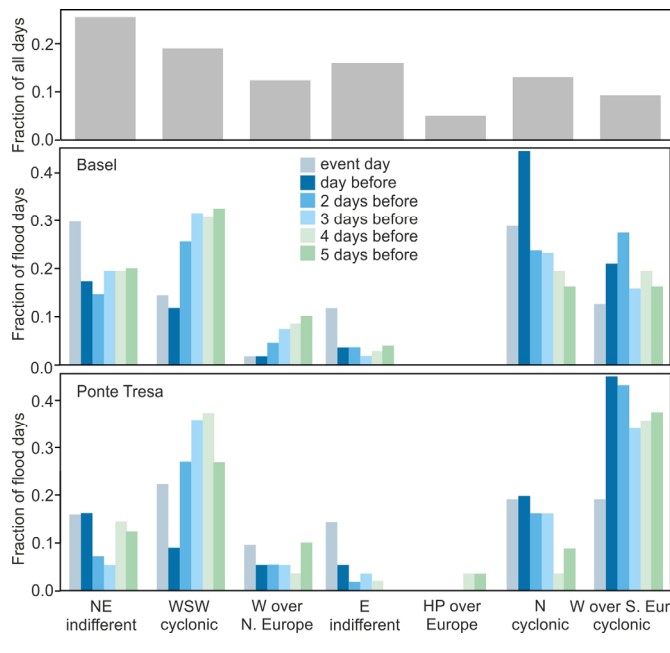


**Fig. 4.** Frequency of CAP7 weather types in the warm season (top). Fraction of flood days occurring during a specific weather types for Basel (middle) and Ponte Tresa (bottom) as well as corresponding series for days 1 to 5 prior to the discharge peak. The figure is based on data from 1901-2000.

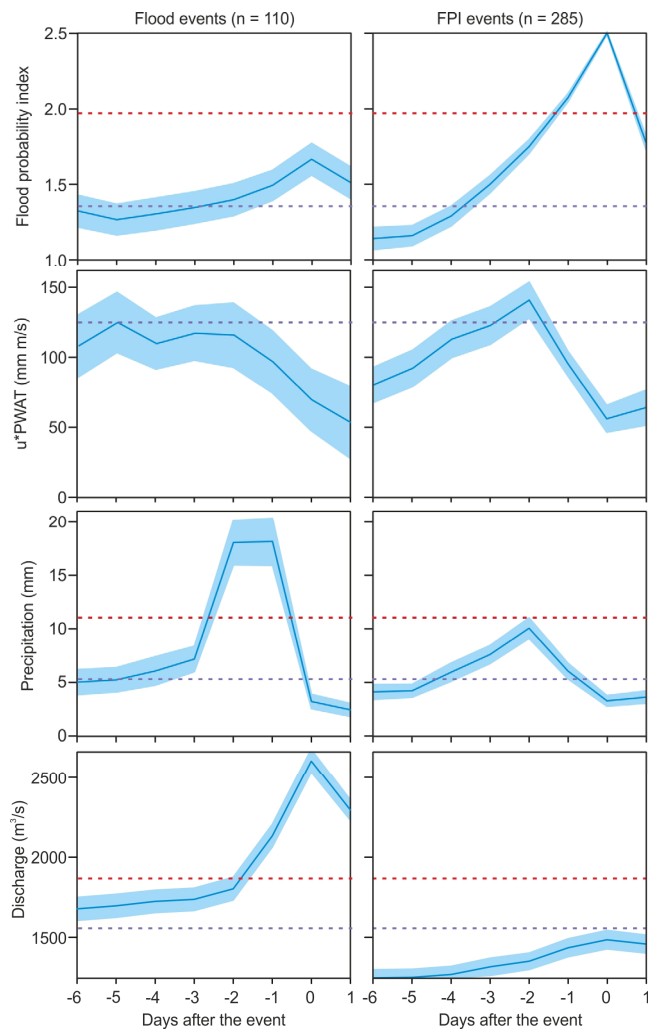


**Fig. 5.** Composites of $FPI_d$, $u_{850hPa}$*PWAT, precipitation, and discharge in Basel for (left) flood events
in Basel and (right) $FPI_d$ events on 1901-2000 for 6 days preceding to 1 day following event day (day
0). Shading indicates two standard deviations. The red and purple dashed lines indicate the 90th and
75th percentile, respectively.

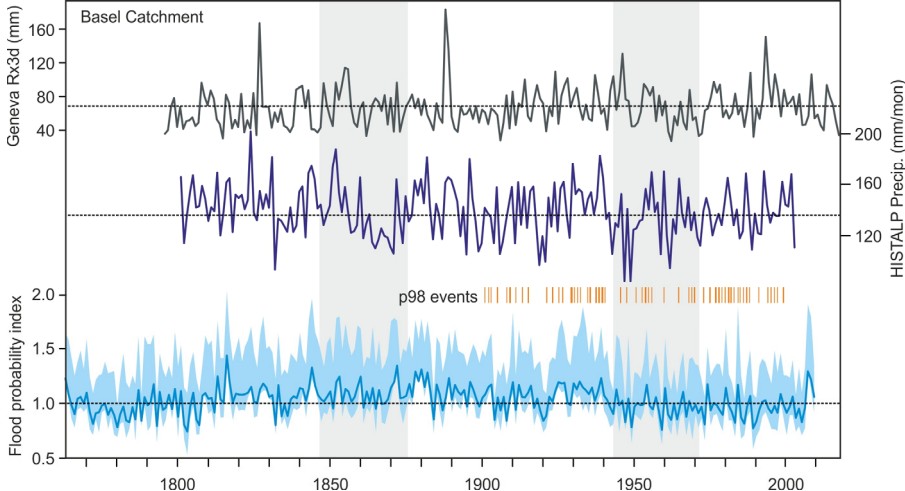


**Fig. 6.** Warm season Rx3d from the station Geneva (top), warm season mean precipitation in
HISTALP for the Rhine catchment (middle) and flood probability index for Basel (bottom, solid
indicates the warm season mean, blue shading indicates the median and 75[th] percentile, respectively).
Dashed lines indicate the 1901-2000 average. Also shown are the peak-over threshold events (p98) of
Basel discharge that were used for calibration. Grey shading denotes the flood-rich period (1847-1876)
and flood-poor period (1943-1972).

648

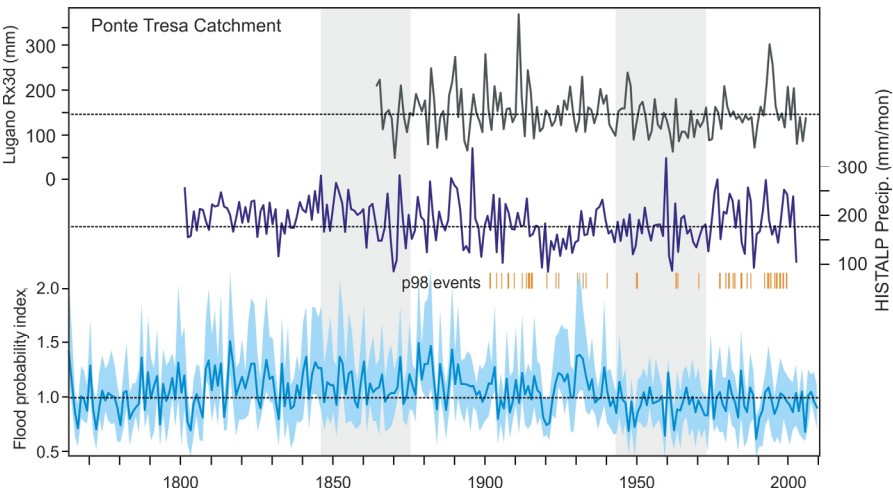

649

**Fig. 7.** Warm season Rx3d from the station Lugano (top), warm season mean precipitation in
HISTALP for the Ponte Tresa catchment (middle) and flood probability index for Ponte Tresa
(bottom, solid indicates the warm season mean, blue shading indicates the median and 75[th] percentile,
respectively). Dashed lines indicate the 1901-2000 average. Also shown are the peak-over threshold
events (p98) of Ponte Tresa discharge that were used for calibration. Grey shading denotes the flood-
rich period (1847-1876) and flood-poor period (1943-1972).

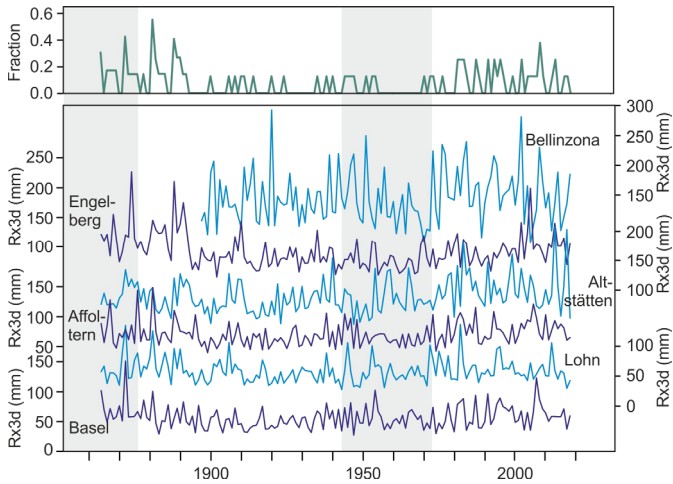


**Fig. 8.** Series of Rx3day for six further stations with long precipitation series (see Fig. 1 for locations).
The top line shows the fraction of these six series plus those of Lugano and Geneva shown in Figs. 5
and 6, exceeding their 95th percentile (based on 1901-2000) in any given year. Grey shading denotes
the flood-rich period (1847-1876) and flood-poor period (1943-1972).

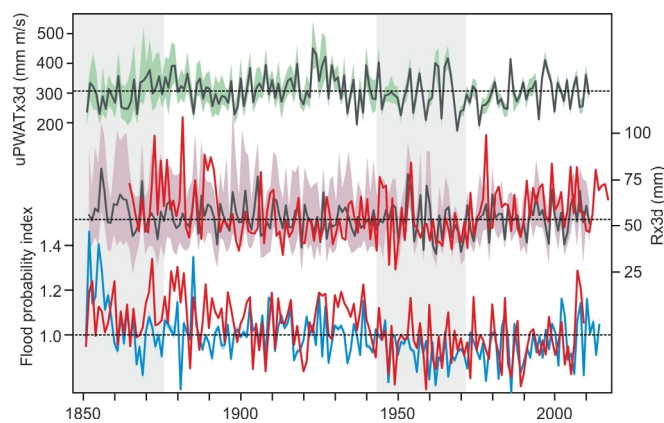


**Fig. 9.** Maximum 3-day average per warm season of (top) eastward moisture transport ($u_{850hPa}$*PWAT)
and (middle) precipitation, both at the grid point 6°E/48°N in 20CRv2c. Bottom: Warm-season mean
FPI index in 20CRv2c. Shading denotes the ensemble range (min. and max.). Red lines show the
corresponding series from observations (Rx3d is calculated form the average of all stations north of
the Alps). Dashed lines indicate the average value for 1901-2000 in 20CRv2c. Grey shadings denote
the flood-rich (1847-1876) and flood-poor (1943-1972) periods, respectively.

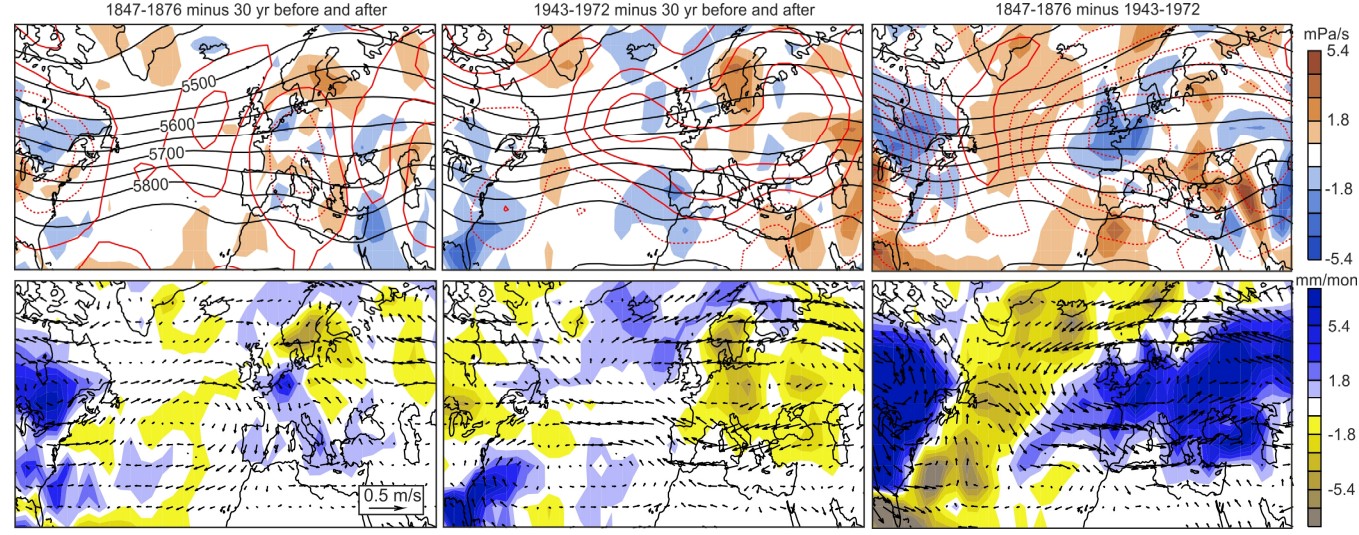


**Fig. 10.** Anomalies of (top) 500 hPa GPH (red contours, 2 gpm spacing symmetric around zero,

negative contours are dashed, black lines indicate the reference period average) and vertical velocity

(colours, lifting is blue) and (bottom) 850 hPa wind and precipitation. Shown are anomalies for the

1847-1876 period (left) and the 1943-1972 period (middle) with respect to the 30 yrs before and after

as well as (right) the difference between the two periods.

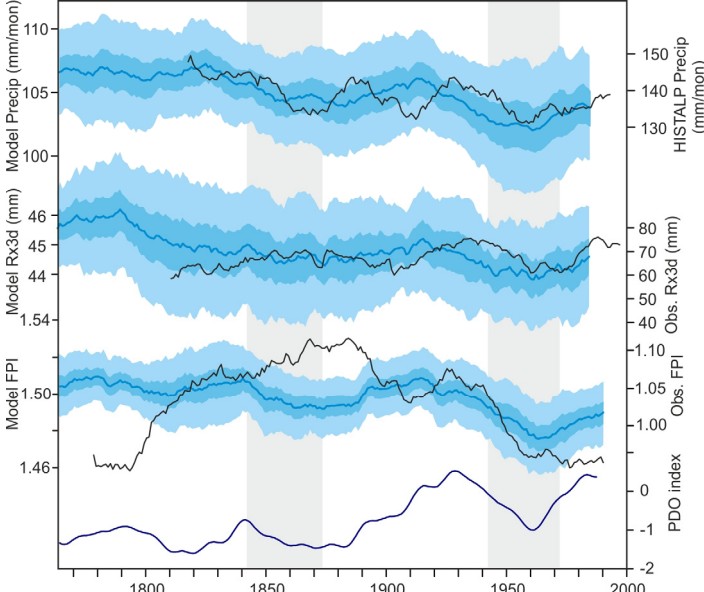

676

**Fig. 11.** CCC400 (left scales) warm season average precipitation (top, note that this series was

detrended based on the corrected member), Rx3day in the warm season at the grid point north of the

Alps (second from top) and flood probability index based on weather types in CCC400 (third from

top). The lowest line shows the PDO index in the model simulations. The solid blue lines show the

ensemble mean of the series smoothed with a 31-yr moving average. Light and dark shadings indicate

the ensemble standard deviation and the 95% confidence interval of the ensemble mean, respectively.

Black lines (right scales) show the corresponding observation-based series. Note the different scales.

Grey shadings denote the flood-rich (1847-1876) and flood-poor (1943-1972) periods, respectively.