# Peer review of "Causes for increased flood frequency in central Europe in the 19th century"

_Climate of the Past, 2019_

## Referee Comment (RC1) · Anonymous Referee #1 · 19 Mar 2019

This paper brings together a wide range of data types in researching the causes of changes in multi-decadal changes in flood frequencies in Europe. This is an interesting topic and I particularly appreciate that it brings together a wide range of data types– observational data, lake level records, paleoclimate reconstructions, and climate model simulations–in addressing the scientific questions.

I have two primary concerns with the present manuscript:

(1) It's not clear to me that the "weather types" and the statistical tools employed here are the best tools for the job. In particular, as discussed at lines 222-228 (and seen in several of the figures), the index that they create has a high rate of false alarms for floods, which the authors attribute to the weather classification scheme only having 7 "types". Doesn't this result imply that a more diverse classification scheme should

be used and that the 7 weather types don't resolve what actually causes floods? It's clear that floods only occur when a cyclone/storm system passes over the area, but the real question is why is it that some storms produce major floods while others do not? That would get at the heart of what really causes floods. A guess based on what I know about the causes of floods in other areas of the world is that the floods may be ultimately caused by something like atmospheric rivers. Atmospheric rivers are "carried" along by mid-latitude cyclones, but many mid-latitude cyclones aren't associated with atmospheric rivers. Something like this could potentially explain the result that the authors see. But if something like this is what's going on, then it seems to me that the authors aren't really looking at the right phenomena. Relatedly, note that also at lines 233-235 you say that the floods in different locations (all within or adjacent to Switzerland) do not occur synchronously, which seems to imply that the causes in each area are very specific and unlikely to be captured by a simple set of 7 types or the even more broad single anomaly pattern shown in Fig 8; furthermore, if floods are extreme or rare events wouldn't you expect their causes to also be rare and not obviously expressed in a set of basic patterns and almost certainly not in a single mean anomaly pattern?

(2) I don't think that the two time scales of analysis are sufficiently bridged. The authors first perform a series of analyses using daily weather data. This time scale makes a lot of sense because that's the time scale over which floods occur. But then the authors jump to a multidecadal analysis using the climate model simulations and the paleoclimate reconstruction, without clearly linking the two very different time scales. It's not obvious to me at all that daily weather patterns should have anything to do with the differences between 30-year climate anomalies. And furthermore, it's not clear that these 30-year climate anomalies have anything to do with the presumably rare mechanism that may actually be underlying the flood events (as suggested by me above in 1). I think the authors need to lay out a very clear and specific set of logical steps that link the underlying daily causes with longer time scales. It's not clear to me, but perhaps the authors are assuming that multidecadal changes in flood frequency implies that there's some driving climate phenomena acting on those scales? That

of course doesn't have to be the case and so the authors may not even have to link the shorter scales to the longer scales. I wouldn't be surprised to see "multidecadal variability" in a rare phenomena (that actually occurs on daily time scales) being due solely to chance. In fact I think that should probably be the null hypothesis here and could explain why the authors find no strong connection to the AMO or PDO.

Minor comments:

I found it somewhat frustrating that the weather types weren't shown in the current manuscript. The explanation in words wasn't really enough for me to clearly see what dynamics were actually going on and I had to pull up the Schwander paper to make sense of what the authors are talking about. Also because the 7 weather types are not shown in this manuscript, it makes it very hard to assess the physical processes going on and how they connect to the model analysis in Fig 8. If the weather types are kept in the analysis here, I would suggest including them among the figures.

For the lake level data, is there reason to be concerned about evaporative effects and the memory of the lake system having an impact on the lake levels? From my reading of the text, it seems like the authors are assuming that the lake levels can be somewhat straightforwardly interpreted as indicating flood events. Maybe they can be, but it's not obvious to me.

Lines 226-228: I don't see the logic of how this sentence follows from what was said previously. Why are 50th and 75th percentiles useful here?

Line 251: I think 1868 is meant here instead of 1968.

Lines 337-339: How does a difference between the two periods imply that there's forced multidecadal variability in the model?

Fig 8: It would be helpful to indicate in the caption where the data and from and which periods are flood-rich vs. flood-poor.

[Figure]

---

## Referee Comment (RC2) · Anonymous Referee #2 · 10 Apr 2019

General comments The author presents an in-depth analysis of multidecadal variations of flood frequency in Switzerland. It is based on long series of discharge and rainfall, weather type reanalysis and climate model simulation. It gives a focus on the Rhine river in Basel and the outlet of Lake Lugano.

The paper is very interesting when it related periods prone to intense or weak rainfall or flood to a flood probability index based on the frequency of weather type. It shows changes in the general meteorological circulation, which can explain the fact that the 19th century was prone to flood event. It gives an interesting contribution to past climate analysis and exemplify that a cold period may have been prone to more frequent floods. It shows clearly that better understanding relations between weather regimes and sea-surface temperature may help research on the assessment of future climate

change.

Specific comments I have only minor remarks on the paper which is well written. It could be improved on the following items.

End of section 2.4 The authors could provide more information on the limits of the reconstruction

Section 2.5 Equation (1) is not clear. I understand that n is relative to a number (as in equation (2), f relates to a frequency). Therefore, I expect to have: wtl = (mtl/nl)/(nt/365) Line 207: "west-southwest cyclonic"

Section 3.1 Line 240: text refers to a flood event in 1882 (Rhine-Basel) which is not visible on Figure 4

Section 3.2 Line 271: give the starting and ending years of the flood rich, flood poor periods Instead of "(p=0.027)", write: "(p-value of the Wilcoxon test: p=0.027)", Line 272: give the starting and ending years of the flood rich, flood poor periods Line 278: "Bibliothèque" Line 284: "over all"

References Lines 143-144: reference of Franke et al. (2017) is missing Line 278: reference of Bibliothèque universelle is missing

---

## Author Comment (AC1) · 1 May 2019

**Reply to reviewer's comments**

This paper brings together a wide range of data types in researching the causes of changes in multi-decadal changes in flood frequencies in Europe. This is an interesting topic and I particularly appreciate that it brings together a wide range of data types–observational data, lake level records, paleoclimate reconstructions, and climate model simulations–in addressing the scientific questions. I have two primary concerns with the present manuscript:

(1) It's not clear to me that the "weather types" and the statistical tools employed here are the best tools for the job. In particular, as discussed at lines 222-228 (and seen in several of the figures), the index that they create has a high rate of false alarms for floods, which the authors attribute to the weather classification scheme only having 7 "types". Doesn't this result imply that a more diverse classification scheme should be used and that the 7 weather types don't resolve what actually causes floods?

Thank you for this comment. Yes, a classification with more types would capture weather types and arguably floods better in the calibration period, but it could not be as well reconstructed back in time. In fact, we tried this. As described in Schwander et al. (2017), we applied our reconstruction technique to classifications with more types, but best results were found for the classification with only 9 types, CAP9, which is one of the official MeteoSwiss classifications. We even had to reduce from 9 to 7 types (hence CAP7), as some of the types could not be well distinguished during the summer season. For our analysis, we opted for the best reconstruction of weather types, one that we trust all the way back to 1763. We will add a sentence on this question to the revised manuscript and refer to Schwander et al. (2017).

It's clear that floods only occur when a cyclone/storm system passes over the area, but the real question is why is it that some storms produce major floods while others do not? That would get at the heart of what really causes floods. A guess based on what I know about the causes of floods in other areas of the world is that the floods may be ultimately caused by something like atmospheric rivers. Atmospheric rivers are "carried" along by mid-latitude cyclones, but many mid-latitude cyclones aren't associated with atmospheric rivers. Something like this could potentially explain the result that the authors see. But if something like this is what's going on, then it seems to me that the authors aren't really looking at the right phenomena.

This is an important point. The reviewer is correct that we miss a diagnostic of moisture transport, which might be one possible reason for higher false alarm rate. In the revised manuscript, we will (1) discuss the role of atmospheric rivers and (2) provide analyses of the Twentieth Century Reanalysis (20CRv2c), for which we have calculated the same weather type classifications, the Flood Probability Indices (FPI), and the same precipitation indices. Additionally, for the Basel catchment, we also calculated a moisture transport diagnostic (u wind at 850 hPa multiplied with precipitable water at a gridpoint to the northwest of the Basel catchment). This shows that moisture transport from the west is already very high several days ahead of the flood event. This finding prompted us to use a slightly longer time window that reaches back to 5 days before the event (and a set of weights that gives less weight to the event day). This new version of the FPI gives almost identical results, but is more consistent with our own findings.

In the revised manuscript, we change to the new index for all analyses (results are visually hardly distinguishable). We further enlarge Figure 3 (which shows the composites and the false alarms rates). We show composites of the FPI, of precipitation (average of all stations north of the Alps), of moisture transport (from 20CRv2c) and of discharge in Basel for (a) selecting flood events in Basel and (b) selecting FPI events. We use the same scales for both selections, which increases comparability.

[Figure]

*Figure: Composites of FPI$_d$, u*PWAT, precipitation, and discharge in Basel for (left) flood events in Basel and (right) FPI$_d$ events on 1901-2000 for 6 days preceding to 1 day following event day (day 0). Shading indicates two standard deviations. The red and purple dashed lines indicate the 90$^{th}$ and 75$^{th}$ percentile, respectively.*

Our analysis shows that moisture transport is a possible, but not the only cause for the false alarm rate. In fact, the moisture transport (at least our diagnostic) is well captured with the FPI. Another cause are the preconditions. Discharge in Basel is already very high a week or more prior to the event. Selecting events according to the daily FPI does not capture this (but an annual mean or 75th percentile of the FPI does capture the frequency of flood-conducive preconditions - this holds part of the answer to the next comment, see below). A third reason for the false alarms is the different sample size of flood events and „FPI events". Even though we use the same threshold (98th percentile of daily data) and declustering, the numbers are very different (110 flood events vs. 285 FPI events) due to the different temporal structure. In the original paper we did not thoroughly explanation the false alarm rate; we will add this discussion to the revised manuscript.

Furthermore, in the revised manuscript, we will show a plot of our 20CRv2c analyses (FPI, moisture flux, precipitation, see below). Due to possible biases, 20CRv2c should not be used alone. For analysing multidecadal variability, we trust our reconstructed weather types more. The figure shows that 20CRv2c has an obvious problem in the FPI in the very early years, but after around 1900 it agrees well. Moisture transport shows similar decadal variability.

[Figure]

*Figure: Maximum 3-day average per warm season of (top) eastward moisture transport ($u_{850hPa}$\*PWAT) and (middle) precipitation, both at the grid point 6°E/48°N in 20CRv2c. Bottom: Warm-season mean FPI index in 20CRv2c. Shading denotes the ensemble range (min. and max.). Red lines show the corresponding series from observations (Rx3d is calculated form the average of all stations north of the Alps). Dashed lines indicate the average value for 1901-2000 in 20CRv2c. Grey shadings denote the flood-rich (1847-1876) and flood-poor (1943-1972) periods, respectively.*

Relatedly, note that also at lines 233-235 you say that the floods in different locations (all within or adjacent to Switzerland) do not occur synchronously, which seems to imply that the causes in each area are very specific and unlikely to be captured by a simple set of 7 types or the even more broad single anomaly pattern shown in Fig 8; furthermore, if floods are extreme or rare events wouldn't you expect their causes to also be rare and not obviously expressed in a set of basic patterns and almost certainly not in a single mean anomaly pattern?

Floods are rare events, but the causes of multidecadal changes in flood occurrence does not have to be rare. Changes in the frequency of rare events can also be the expression of a change that affects a larger part of (or the entire) the distribution. In addition, as noted above, preconditions (and thus more than just one extreme) also matter. The FPI shows that changes in high percentiles are closely mirrored by changes in the mean. For instance, the correlations of the 75th and 90th percentile with the mean are 0.92 and 0.77 for the Basel catchment and 0.95 and 0.91 for the Ponte Tresa catchment. We will add this information to the revised manuscript and state more clearly that changes in flood events may partly be the expression of a change in the upper part of the distribution, plus they depend on preconditions.

(2) I don't think that the two time scales of analysis are sufficiently bridged. The authors first perform a series of analyses using daily weather data. This time scale makes a lot of sense because that's the time scale over which floods occur. But then the authors jump to a multidecadal analysis using the climate model simulations and the paleoclimate reconstruction, without clearly linking the two very different time scales. It's not obvious to me at all that daily weather patterns should have anything to do with the differences between 30-year climate anomalies. And furthermore, it's not clear that these 30-year climate anomalies have anything to do with the presumably rare mechanism that may actually be underlying the flood events (as suggested by me above in 1). I think the authors need to lay out a very clear and specific set of logical steps that link the underlying daily causes with longer time scales. It's not clear to me, but perhaps the authors are assuming that multidecadal changes in flood frequency implies that there's some driving climate phenomena acting on those scales? That of course doesn't have to be the case and so the authors may not even have to link the shorter scales to the longer scales. I wouldn't be surprised to see "multidecadal variability" in a rare phenomena (that actually occurs on daily time scales) being due solely to chance. In fact I think that should probably be the null hypothesis here and could explain why the authors find no strong connection to the AMO or PDO.

Again a helpful comment. We were switching too rapidly to the decadal scale and were not explicit as to the underlying assumptions. In fact, we do not imply a driving mechanism at the decadal scale. As we state at the end, we think that a lot of this multidecadal variability is random or atmospheric, and this is our null-hypothesis (only the 20th century drop in flood occurrence might have been forced; but overall results are not statistically significant). This is nevertheless important as such variability is likely to occur also in a future climate. We will more clearly state in the revised manuscript that multidecadal variability might occur by chance. The introduction to Section 3.3, where the change to the decadal scale occurs, will be rewritten to make this change more explicit. We will refer again to the notion that changes in extremes might be the expression of changes in the upper part of the distribution, that we in fact diagnose such changes in the distribution in the FPI, and that we now analyse the relation of this to decadal scale changes in the mean circulation.

Remember that the starting point of the paper was the perceived multidecadal variability of flood occurrence. The goal of our paper was to clarify whether this was real or whether it was only perceived, and if it was real, whether we can track the signal back to precipitation changes, and if we can trace it back to precipitation, whether this is related to changes in the frequency of weather types and finally to changes in the mean state of atmospheric circulation. We can demonstrate all of this, except that we cannot relate the multidecadal variability to forcings in our model. So, the atmospheric causes might have been largely unforced (or our model might not adequately represent the mechanisms). We will make this argumentation more clear in the revised manuscript.

*Minor comments:*

I found it somewhat frustrating that the weather types weren't shown in the current manuscript. The explanation in words wasn't really enough for me to clearly see what dynamics were actually going on and I had to pull up the Schwander paper to make sense of what the authors are talking about. Also because the 7 weather types are not shown in this manuscript, it makes it very hard to assess the physical processes going on and how they connect to the model analysis in Fig 8. If the weather types are kept in the analysis here, I would suggest including them among the figures.

Thanks – we will include a figure with the circulation patterns for the 7 weather types.

For the lake level data, is there reason to be concerned about evaporative effects and the memory of the lake system having an impact on the lake levels? From my reading of the text, it seems like the authors are assuming that the lake levels can be somewhat straightforwardly interpreted as indicating flood events. Maybe they can be, but it's not obvious to me.

Yes, lake floods are not quite the same as river floods, as they more strongly depend on the antecedent lake level, which may have a long memory. We add a sentence on that.

Lines 226-228: I don't see the logic of how this sentence follows from what was said previously. Why are 50th and 75th percentiles useful here?

It is true that the argument was more implicit than explicit. Because of the false alarm rate discussed above, there is not much interannual variability in high percentiles of the FPI (flood-conducive cyclone passages occur almost every summer); the index „saturates". The 75th percentile captures the upper part of the distribution, mean and 50th percentile capture the central tendency (and thus how the signal translates from the mean to the extremes). We will justify this more explicitly in the revised manuscript.

Line 251: I think 1868 is meant here instead of 1968.

Thanks

Lines 337-339: How does a difference between the two periods imply that there's forced multidecadal variability in the model?

„Forced" refers to all boundary conditions used in the model simulations. A significant difference (p = 0.005) between two periods in the ensemble mean suggests that it is the boundary conditions (of which SSTs are arguably the most important) and not the ensemble variability. Note that we are looking at an ensemble mean of 30 simulations over 30 or 60 years (thus 900 model years against 1800 climatology years or against another 900 years). Unforced variability averages out quite efficiently.

Fig 8: It would be helpful to indicate in the caption where the data and from and which periods are flood-rich vs. flood-poor.

Thanks, we add that to each plot.

---

## Author Comment (AC2) · 1 May 2019

**Reply to reviewer's comments**

General comments The author presents an in-depth analysis of multidecadal variations of flood frequency in Switzerland. It is based on long series of discharge and rainfall, weather type reanalysis and climate model simulation. It gives a focus on the Rhine river in Basel and the outlet of Lake Lugano.

The paper is very interesting when it related periods prone to intense or weak rainfall or flood to a flood probability index based on the frequency of weather type. It shows changes in the general meteorological circulation, which can explain the fact that the 19th century was prone to flood event. It gives an interesting contribution to past climate analysis and exemplify that a cold period may have been prone to more frequent floods. It shows clearly that better understanding relations between weather regimes and sea-surface temperature may help research on the assessment of future climate change.

Specific comments I have only minor remarks on the paper which is well written. It could be improved on the following items.

End of section 2.4 The authors could provide more information on the limits of the reconstruction.

We will add a sentence on the reconstruction uncertainty. For our analyses we chose the classification which can be reconstructed best (CAP7). After 1810, the probability of each day to be attributed to the right class is higher than 80%, after 1860 it is higher than 85%. Using a classification with more classes might caputre flood events better, but could not be well reconstructed.

Section 2.5 Equation (1) is not clear. I understand that n is relative to a number (as in equation (2), f relates to a frequency). Therefore, I expect to have: wtl = (mtl/nl)/(nt/365)

This was not very clear; the choice of the variable names $n$ and $f$ was misleading. We will be more specific and state more clearly that we use absolute frequency (counts) and not relative frequencies. We will replace $f$ with $n$ to make this clear.

Line 207: "west-southwest cyclonic"

Thanks.

Section 3.1 Line 240: text refers to a flood event in 1882 (Rhine-Basel) which is not visible on Figure 4.

In the revised paper we will add a small arrow to the figure.

Section 3.2 Line 271: give the starting and ending years of the flood rich, flood poor periods Instead of "(p=0.027)", write: "(p-value of the Wilcoxon test: p=0.027)"

Thanks.

Line 272: give the starting and ending years of the flood rich, flood poor periods

Thanks.

Line 278: "Bibliothèque"

Thanks.

Line 284: "over all"

Thanks.

References

Lines 143-144: reference of Franke et al. (2017) is missing

Line 278: reference of Bibliothèque universelle is missing